# Scalable Detection of Undiagnosed ILD in Population Screening: A Multi-Cohort Study using 3D Foundation Models

**Niccolò McConnell**[1,2]                          NICCOLO.MCCONNELL@UCL.AC.UK
**Mehran Azimbagirad**[1]                          M.AZIMBAGIRAD@UCL.AC.UK
**Daryl O. Cheng**[1,4]                            DARYL.CHENG@UCL.AC.UK
**Daisuke Yamada**[1]                              D.YAMADA@UCL.AC.UK
**Ryoko Egashira**[1]                              R.EGASHIRA@UCL.AC.UK
**Robert Chapman**[1]                              ROB.CHAPMAN@UCL.AC.UK
**John McCabe**[1,4]                               J.MCCABE@UCL.AC.UK
**Shanshan Wang**[1]                               SHANSHAN.WANG.23@UCL.AC.UK
**David Lynch**[7]                                 LYNCHD@NJHEALTH.ORG
**Greg Kinney**[6]                                 GREG.KINNEY@CUANSCHUTZ.EDU
**Pardeep Vasudev**[1,2]                           PARDEEP.VASUDEV.19@UCL.AC.UK
**SUMMIT Consortium**[*]
**Paul Taylor**[2]                                 P.TAYLOR@UCL.AC.UK
**Daniel C. Alexander**[1]                         D.ALEXANDER@UCL.AC.UK
**Sam M. Janes**[3]                                S.JANES@UCL.AC.UK
**Joseph Jacob**[1,5]                              J.JACOB@UCL.AC.UK

[1] *Satsuma Lab, Hawkes Institute, University College London, UK*

[2] *Institute of Health Informatics, University College London, UK*

[3] *Lungs for Living Research Centre, University College London, UK*

[4] *Dept of Radiology, University College London Hospitals NHS Trust, UK*

[5] *UCL Respiratory, University College London, UK*

[6] *Colorado School of Public Health, University of Colorado Anschutz, USA*

[7] *National Jewish Health, Dept. Radiology, Denver, USA*

[*] *Full consortium list provided in the Acknowledgments.*

**Editors:** Accepted for publication at MIDL 2026

## Abstract

Undiagnosed interstitial lung disease (UILD), an early form of lung fibrosis, is increasingly detected in population-based low-dose computed tomography (LDCT) screening but remains systematically under-reported due to its subtle appearance. We developed and validated a foundation-model-augmented deep learning system for UILD detection across two of the largest thoracic CT cohorts worldwide: SUMMIT, the UK's largest LDCT screening study (>11,000 scans), and COPDGene, a multi-centre US cohort spanning 21 scanners and >8,800 scans. We propose ViT-3D-TE, a multi-token 3D Vision Transformer designed to preserve both high-frequency focal texture and diffuse parenchymal change through CLS, MAX, and AVG token fusion. The model was initialised with TANGERINE, an open-source 3D masked autoencoder pretrained on 98,000 full-volume LDCT scans, providing volumetric priors essential for stable optimisation. ViT-3D-TE was trained solely on SUMMIT and evaluated on COPDGene without domain adaptation, and achieved strong

performance (AUROC 0.9805, AUPRC 0.7699 internal; AUROC 0.9705, AUPRC 0.6170 external), representing $17\times$ and $25\times$ improvements over random baselines at clinically realistic cohort prevalences (4.6% and 2.5%). We further introduce ConvNeXt-2.5-MIL, a slice-based 2.5D alternative that performs competitively without relying on 3D foundation model pretraining. Together, these results provide, to our knowledge, the largest real-world validation to date of deep learning for UILD detection and demonstrate that foundation-model-enhanced 3D Transformers offer a practical and scalable pathway for integrating UILD detection into national LDCT screening workflows.

**Keywords:** Foundation Model, Large-scale CT data, UILD, ViT, ConvNeXt.

## 1. Introduction

Interstitial lung disease (ILD) comprises a heterogeneous group of disorders characterised by inflammation, fibrosis, and progressive architectural distortion of the lung parenchyma. Within this spectrum, recent screening-cohort studies have identified an under-recognised subgroup with definite fibrotic changes on CT but no prior ILD diagnosis. This entity, termed undiagnosed interstitial lung disease (UILD), is defined radiologically by faint reticulation coexisting with early traction bronchiolectasis across three or more lung lobes, falling below existing diagnostic thresholds yet remaining prognostically important (Hunninghake et al., 2022; Hatabu et al., 2020). Individuals with UILD have significantly higher risks of respiratory hospitalisation and mortality than CT-normal peers, even after adjusting for age and smoking history (Putman et al., 2016; Sanders et al., 2023). UILD is not rare. Across large lung cancer screening programmes, early fibrotic abnormalities are seen in approximately 4–8% of older adults (Jin et al., 2013; Brown et al., 2019), but most cases remain unreported because findings are low-contrast, spatially heterogeneous, and often dismissed as benign age-related change (Hatabu et al., 2020; Hunninghake et al., 2022).

The rapid expansion of population low-dose CT (LDCT) programmes heightens the urgency for automated early fibrosis detection. Initiatives such as the UK Targeted Lung Health Check and comparable international efforts now generate millions of volumetric scans annually (de Koning et al., 2020; Team, 2011; Cancer Research UK, 2024). Although designed primarily for cancer detection, these programmes offer an opportunity to identify incipient interstitial lung disease at a stage when lung function is often preserved and intervention may still alter disease trajectory (de Mattos et al., 2022). However, integrating ILD assessment into routine screening faces major capacity constraints: subtle UILD patterns are easily overlooked by generalist radiology readers, inter-reader agreement is modest, and expert review of every LDCT scan is infeasible at national scale (The Royal College of Radiologists, 2023). These factors create a compelling need for robust, scalable AI systems capable of flagging fibrotic abnormalities.

Despite progress in deep learning for fibrosis staging and ILD subtype classification, automated detection of UILD in screening populations remains limited. Early fibrotic changes often appear as fine textural deviations in the peripheral lung, sometimes spanning only a few slices. These characteristics make UILD particularly challenging for standard 3D Vision Transformers (ViTs) (Dosovitskiy, 2020), which rely on deep semantic abstractions formed after extensive patch mixing. This process can oversmooth high-frequency parenchymal texture cues–such as early reticulation or traction change–that radiologists depend on for diagnosis, leading to false negatives and unstable decision boundaries.

To address these limitations, we propose ViT-3D-TE, a multi-token fusion 3D Vision Transformer designed to better preserve radiologically meaningful texture. The architecture integrates complementary representations from the final encoder block: (i) a global [CLS] token summarising overall thoracic context and lobar distribution; (ii) a MAX-pooled embedding that encourages sensitivity to focal, high-saliency abnormalities; and (iii) an AVG-pooled embedding that encourages retention of diffuse, low-amplitude parenchymal change. This token-ensemble design is motivated by radiological reasoning, where both focal traction change and subtle distributed reticulation contribute to early disease assessment.

A second cornerstone of our approach is TANGERINE (McConnell et al., 2025), a 3D masked-autoencoder foundation model pretrained on more than 98,000 full-volume chest CT scans. TANGERINE provides robust anatomical priors and stabilises full-volume ViT optimisation, which would otherwise be brittle under the substantial class imbalance and subtle pathology characteristic of UILD.

**Contributions.** This work makes four primary contributions:

1. We develop a deep learning system for detecting undiagnosed interstitial lung disease (UILD) under clinically realistic disease prevalence, trained and internally validated on the SUMMIT lung cancer screening cohort and externally validated on COPDGene.

2. We introduce *ViT-3D-TE*, a token-ensemble 3D Vision Transformer that aims to preserve high-frequency focal texture cues alongside global context, mitigating the oversmoothing limitations of standard 3D Transformers.

3. We demonstrate the critical role of large-scale 3D foundation model pretraining in this setting, showing that the TANGERINE encoder substantially improves performance over convolutional baselines and unpretrained Transformers.

4. We propose ConvNeXt-2.5-MIL, a 2.5D slice-based MIL architecture with gated attention and cranio-caudal positional encoding that achieves strong performance in the absence of 3D foundation model pretraining.

## 2. Related Works

Deep learning has been applied to a range of ILD diagnosis tasks in specialist cohorts, primarily using high-resolution CT (HRCT). Prior work includes models for idiopathic pulmonary fibrosis (IPF) diagnosis and acute exacerbation detection, pattern-subtype classifiers using 3D CNNs or radiomics, and multimodal frameworks that integrate CT with clinical and pathological data (Yu et al., 2023; Huang et al., 2024; Zhang et al., 2025; Kumarganesh et al., 2025; Baba et al., 2025). Across these HRCT clinic studies, models are trained on hundreds of patients from tertiary ILD centres, often incorporate additional clinical priors, and address tasks such as IPF diagnosis, acute exacerbation detection, or pattern subtyping in already-diagnosed ILD.

Most closely related to our setting, Chen et al. evaluated ScreenDx, an FDA-cleared proprietary deep learning tool, for automated detection of early or under-diagnosed ILD in a COPDGene sub-cohort (2,280 subjects; 28 true-positive ILD cases), reporting a sensitivity of 84.8% and specificity of 98.0% at a single pre-selected operating point (Chen et al.,

Table 1: Summary of datasets used for internal development (SUMMIT) and external validation (COPDGene). Counts for UILD and non-UILD are shown per split where applicable.

| Dataset | Total Scans | UILD (n) | Non-UILD (n) | Train | Validation | Test | Notes |
|---|---|---|---|---|---|---|---|
| **SUMMIT** | 11,179 | Train: 275
Val: 51
Test: 102 | Train: 7,550
Val: 1,066
Test: 2,135 | 7,825 | 1,117 | 2,237 | UK LDCT screening cohort; labels from 3 thoracic radiologists |
| **COPDGene** | 8,874 | 220 | 8,654 | – | – | 8,874 | External inference-only evaluation; no fine-tuning performed |

2025). However, ScreenDx is a closed-source black-box system with no publicly available architectural details, is assessed on a relatively small, retrospectively constructed sample with few positive cases and case-enriched prevalence, and is evaluated without threshold-free or prevalence-aware metrics (e.g. AUROC, AUPRC), limiting its direct applicability to national screening workflows. We instead target UILD in population low-dose CT screening, use full-volume LDCT alone as input, initialise a 3D Transformer with a large-scale foundation model, and provide, to the best of our knowledge, the first deep learning based large-scale multi-cohort study of cross-cohort generalisation for UILD detection, analysing more than 20,000 screening LDCT scans and evaluating performance on over 10,000 scans at clinically realistic prevalence.

## 3. Methods

### 3.1. Datasets

#### 3.1.1. SUMMIT Cohort (Internal Development & Evaluation)

The internal dataset was derived from the SUMMIT Study (Dickson et al., 2023; Bhamani et al., 2025), a UK lung cancer screening cohort of at-risk adults aged 55–77 years undergoing baseline low-dose CT (LDCT) with a standardised protocol. Three fellowship-trained thoracic radiologists independently reviewed each scan and assigned a binary label indicating definite undiagnosed interstitial lung disease (UILD) by expert consensus. While initial inter-reader disagreement was not explicitly stratified, borderline and marginal cases are inherently included in both training and evaluation through this consensus process. Train/validation/test splits were performed at the patient level (70%/10%/20%), with UILD and non-UILD counts summarised in Table 1.

#### 3.1.2. COPDGene Cohort (External Validation)

External generalisation was assessed using COPDGene (Regan et al., 2011), a multi-centre US cohort of smokers and ex-smokers imaged across 21 scanners with heterogeneous acquisition parameters. We included 8,874 baseline inspiratory scans. The same radiological criteria as in SUMMIT were used to assign UILD labels. No model was fine-tuned, calibrated, or adapted on COPDGene; all evaluations were performed strictly in inference mode (Table 1).

### 3.2. Task Definition

We formulate UILD detection as a binary classification task distinguishing UILD (label = 1) from a composite Non-UILD class (label = 0). The negative class was intentionally broad, including radiologically normal controls and individuals with other abnormalities and comorbidities (e.g., non-fibrotic ILA, bronchiectasis, COPD) that do not meet criteria for UILD. This design imposes a stringent discriminative objective, requiring the model to identify prognostic patterns of early fibrotic ILD while rejecting common parenchymal mimics.

### 3.3. CT Preprocessing

All CT scans underwent a standardised preprocessing pipeline to ensure consistent spatial alignment and radiometric comparability across scanners:

- **Resampling:** All volumes were resampled to $1.35\,\text{mm}$ isotropic spacing.

- **Lung segmentation:** Automated whole-lung masks generated using TotalSegmentator (Wasserthal et al., 2023).

- **Lung-centric normalisation:** Each volume was rigidly translated and embedded into a $256^3$ voxel grid, centring the segmented lung fields.

- **HU clipping:** Intensities clipped to $[-1200, 800]\,\text{HU}$.

- **Intensity scaling:** Voxels min–max normalised to the range [0,1].

### 3.4. Model Architectures

#### 3.4.1. RESNET-VANILLA (3D, NON-PRETRAINED)

A standard 3D ResNet-50 (He et al., 2016) was trained from scratch using He initialisation. Operating directly on the full $256^3$ volume, this model provides a baseline for fully supervised learning without domain-specific priors.

#### 3.4.2. MED3D-RESNET (3D, MEDICAL-DOMAIN PRETRAINED)

We fine-tuned a ResNet-50 encoder pretrained via Med3D (Chen et al., 2019) on multi-organ segmentation tasks. Volumes were normalised using per-volume standardisation to match Med3D conventions. This model investigates the effect of generic 3D medical pretraining compared with training from scratch.

#### 3.4.3. CONVNEXT-2.5-MIL (2.5D MULTIPLE-INSTANCE LEARNING)

Unlike the fully 3D models, ConvNeXt-2.5-MIL (Woo et al., 2023; Ilse et al., 2018) operates in a 2.5D regime (Fig. 1), decomposing each CT volume into $N$ uniformly spaced axial slices (e.g., $N = 128$), each treated as an instance within a MIL bag. This reduces spatial dimensionality while retaining high in-plane resolution. We selected $N = 128$ slices as a balance between anatomical coverage and computational efficiency; preliminary experiments

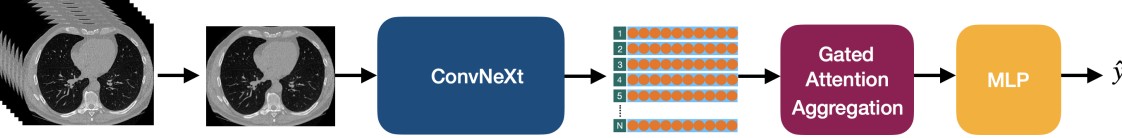

Figure 1: Overview of the proposed ConvNeXt-2.5-MIL architecture. Each CT volume is decomposed into $N$ axial slices processed by a ConvNeXt-V2-Tiny backbone; slice embeddings, augmented with cranio-caudal positional encoding, are assigned learned importance weights by gated attention and combined through weighted-sum pooling into a patient-level representation for UILD prediction.

indicated limited sensitivity to this choice within a reasonable range (approximately $\pm20$ slices).

For each slice, features are extracted using a ConvNeXt-V2-Tiny backbone initialised with FCMAE weights (Woo et al., 2023; He et al., 2022). These features are globally average pooled, passed through per-slice LayerNorm, and concatenated with a normalised cranio-caudal positional encoding ($0 = $ apex, $1 = $ lung bases). A gated attention module computes learned importance weights over these position-aware slice embeddings, which are then combined through weighted-sum pooling into a patient-level representation for binary classification. The positional encoding is clinically motivated by the basal predominance of early interstitial abnormalities.

*Novel aspects:* Our implementation combines (i) slice-wise LayerNorm to mitigate inter-slice heterogeneity, (ii) explicit cranio-caudal positional encoding, and (iii) a modern ConvNeXt-V2 backbone pretrained via masked autoencoding. To our knowledge, this combination has not been applied to ILD or UILD classification and provides a texture-sensitive alternative to full-volume 3D Transformers.

### 3.4.4. ViT-C, ViT-HF, and ViT-NP

We adapted a 3D ViT-Large architecture to full-volume CT. Each $256^3$ volume was partitioned into $16^3$ patches, embedded into 1024-dimensional vectors, and augmented with fixed 3D sinusoidal positional encodings. In the CLS-only baseline (ViT-C), a learnable CLS token is prepended and its final representation alone is used for prediction. To evaluate multi-scale aggregation, ViT-HF concatenated MAX-pooled intermediate block features (layers 8 and 16) with the final CLS token. To isolate the effect of pretraining, ViT-NP used the same triplet-pooling architecture as our proposed model but was trained from scratch with He initialisation.

### 3.4.5. ViT-3D-TE (Final Proposed Model)

The proposed ViT-3D-TE (Fig. 2) extends the CLS-only ViT with triplet token aggregation from the final encoder layer: (i) the CLS token (global thoracic context), (ii) a spatial MAX-pooled embedding that encourages sensitivity to focal abnormalities, and (iii) an AVG-pooled embedding that encourages retention of diffuse low-contrast changes. Standard ViTs progressively mix and homogenise patch embeddings at depth, potentially attenuating

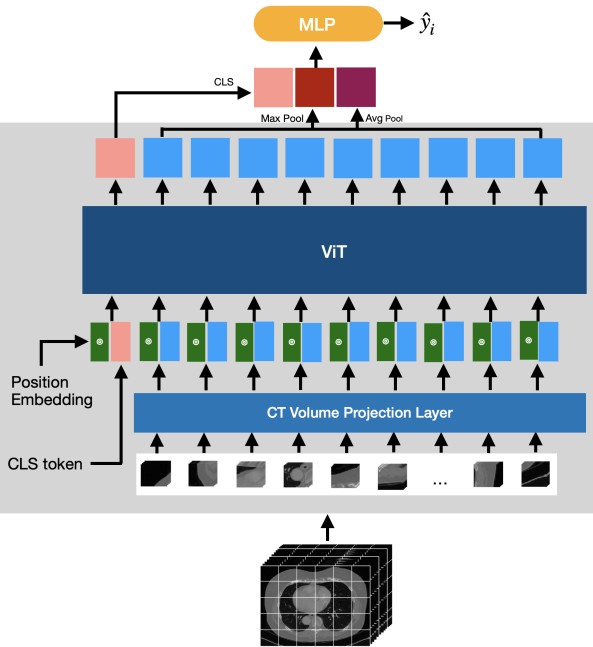

Figure 2: Overview of the proposed ViT-3D-TE architecture. A 3D CT volume is partitioned into patches and encoded by a 3D Vision Transformer; the final CLS, MAX-pooled, and AVG-pooled representations are concatenated into a token-ensemble vector and passed to an MLP for UILD prediction.

such high-frequency parenchymal texture. By aggregating MAX and AVG pooled representations alongside the CLS token, ViT-3D-TE is encouraged to retain both focal and diffuse cues. The three 1024-dimensional vectors are concatenated into a 3072-dimensional representation, normalised, and passed through a lightweight MLP head.

We emphasise that ViT-3D-TE is not intended as a new class of Vision Transformer, nor as a generic pooling or multi-scale aggregation strategy. Rather, it represents a task-specific architectural adaptation motivated by radiological reasoning. Unlike intermediate pooling or post-hoc feature aggregation, the MAX- and AVG-pooled representations are extracted from the final encoder layer and are jointly supervised alongside the CLS token. This final-layer multi-token supervision explicitly encourages the retention of complementary signal types, including global thoracic context, focal high-saliency abnormalities, and diffuse low-contrast parenchymal change, that are otherwise progressively homogenised in deep 3D Vision Transformers.

### 3.4.6. Note on TANGERINE Pretraining

All ViT variants except ViT-NP were initialised with the encoder from *TANGERINE* (Mc-Connell et al., 2025), a 3D masked autoencoder (He et al., 2022) pretrained on more than 98,000 full-volume LDCT scans from 28 datasets across eight countries. TANGERINE provides volumetric priors on thoracic anatomy, reconstruction variability, and parenchy-

mal texture, offering a strong inductive bias and improving label efficiency, stability, and cross-cohort generalisation for subtle entities such as UILD.

### 3.5. Training Strategy

All models were trained end-to-end with AdamW (weight decay $10^{-4}$). Pretrained models used a base learning rate of $5 \times 10^{-5}$; non-pretrained models used $1 \times 10^{-3}$. A five-epoch warm-up preceded cosine decay. Transformer models used layer-wise learning-rate decay (ratio = 0.8). Effective batch size was 32 using gradient accumulation. Augmentations included Gaussian noise, smoothing, and contrast jitter. Given the low UILD prevalence, we applied BCEWithLogitsLoss with a positive-class weight reflecting the empirical training-set ratio. All models were trained for 75 epochs, with the checkpoint used for inference selected according to the epoch with highest validation AUPRC.

### 3.6. Evaluation Protocol and Statistical Analysis

Internal performance is reported on the held-out SUMMIT test set; models were trained on the SUMMIT training split and tuned on the validation set. External evaluation was performed on all COPDGene scans in strict inference mode, without fine-tuning, domain adaptation, calibration, or threshold adjustment.

We report AUROC and AUPRC as threshold-free metrics, and sensitivity/ specificity at a single operating point. The threshold was chosen on the SUMMIT validation set as the largest value achieving sensitivity $\geq 0.90$ and then fixed for both SUMMIT (test) and COPDGene, reflecting clinical preferences for high sensitivity in early-detection settings.

Ninety-five percent confidence intervals were computed via patient-level non-parametric bootstrapping with 5,000 resamples. To assess differences between models, we used paired bootstrap testing: for each resample, AUROC and AUPRC were computed for all models, paired differences versus ViT-3D-TE were obtained, and two-sided $p$-values were derived from the empirical distribution of differences.

### 3.7. Implementation

Models were implemented in PyTorch 1.13 with timm 0.9 backbones and trained on NVIDIA A6000 GPUs. For inference efficiency, we measured single-case runtime and peak GPU memory on an A6000: ViT-3D-TE processed a $256^3$ volume in $607\,\text{ms}$ (1.47 GB), while ConvNeXt-2.5-MIL, using 128 axial slices of $256 \times 256$, achieved $368\,\text{ms}$ per scan (3.05 GB). Both models operate comfortably within modern GPU constraints, with ConvNeXt-2.5-MIL offering lower latency and ViT-3D-TE lower VRAM demand. Code is available at https://github.com/niccolo246/UILD-detection-deep-learning.

## 4. Results

### 4.1. Internal Validation - SUMMIT

We first evaluated all models on the SUMMIT internal test set, where training and evaluation data originate from the same screening population (Table 2, Fig. 4a–b). ViT-3D-TE

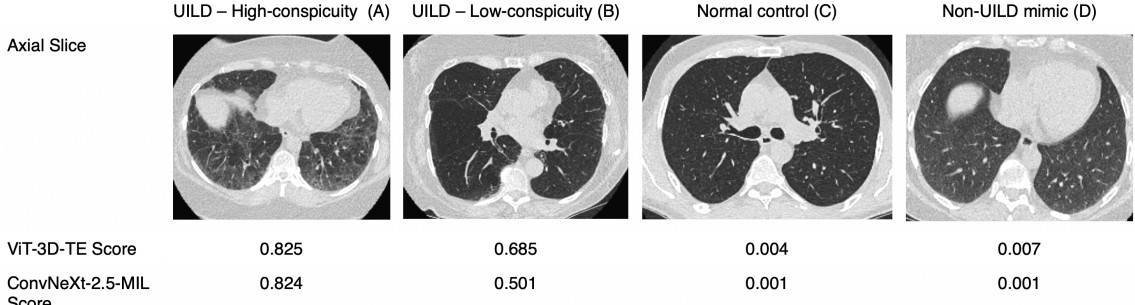

Figure 3: Qualitative examples illustrating model behaviour across representative LDCT screening cases. Axial CT slices are shown for (A) a UILD case with visually conspicuous fibrotic features, (B) a UILD case with low visual conspicuity, (C) normal control, and (D) a non-UILD mimic with diffuse ground-glass-like attenuation. For each case, predicted UILD probabilities from ViT-3D-TE and ConvNeXt-2.5-MIL are reported. Images are cropped to the lung fields and displayed using a consistent lung window (HU) for visual clarity; all slices correspond to the $256^3$ downsampled volumes used as model input. Examples are intended to illustrate typical model behaviour and confounding patterns rather than disease severity.

achieved the strongest performance, with AUROC 0.9805 and AUPRC 0.7699, outperforming all comparison models. ConvNeXt-2.5-MIL was the closest competitor (AUROC 0.9712), but remained significantly below ViT-3D-TE in AUROC ($\Delta$AUROC $= -0.0094$, $p = 0.004$), while AUPRC differences did not reach significance ($p = 0.082$).

Performance dropped substantially for the 3D convolutional baselines (ResNet-vanilla AUROC 0.77; Med3D-ResNet AUROC 0.56), both significantly worse than ViT-3D-TE across AUROC and AUPRC (all $p < 10^{-4}$). ROC and PR curves (Fig. 4a–b) show ViT-3D-TE maintaining clear separation across thresholds, ConvNeXt-2.5-MIL remaining competitive but saturating earlier in the low-precision regime, and the convolutional models exhibiting flattened curves consistent with limited discriminative capacity at high-sensitivity operating points.

Qualitative examples in Fig. 3 provide case-level insight into model behaviour across high and low UILD conspicuity, and confounding screening scenarios, illustrating both successful detections and the types of ambiguous cases that challenge automated UILD classification. As expected in a population screening setting, both models occasionally assign elevated probabilities to non-UILD fibrotic mimics, such as atelectasis or emphysema with superimposed ground-glass opacities (Appendix Fig. 5), reflecting known challenges in distinguishing early fibrotic change from confounding parenchymal patterns.

## 4.2. External validation - COPDGene

We next evaluated cross-cohort generalisation on the COPDGene external test set, where training (SUMMIT) and evaluation populations differ in scanner type, acquisition protocol,

Table 2: Performance comparison on the SUMMIT internal test set. AUROC and AUPRC are reported with 95% bootstrap confidence intervals.

| Model | AUROC (95% CI) | AUPRC (95% CI) | Sensitivity | Specificity |
|---|---|---|---|---|
| ViT-3D-TE | 0.9805 [0.9709, 0.9875] | 0.7699 [0.6918, 0.8312] | 0.9706 | 0.9091 |
| ConvNeXt-2.5-MIL | 0.9712 [0.9584, 0.9815] | 0.7353 [0.6517, 0.8046] | 0.9216 | 0.9115 |
| Med3D-ResNet | 0.5587 [0.5050, 0.6107] | 0.0645 [0.0444, 0.0952] | 0.9020 | 0.1719 |
| ResNet-vanilla | 0.7720 [0.7403, 0.8037] | 0.1158 [0.0820, 0.1556] | 0.9216 | 0.5541 |

Table 3: Paired bootstrap significance testing on the SUMMIT internal test set. Baseline model is marked with †. $\Delta$ values represent paired differences relative to the baseline.

| Model | $\Delta$AUROC (95% CI) | p(AUROC) | $\Delta$AUPRC (95% CI) | p(AUPRC) |
|---|---|---|---|---|
| ViT-3D-TE[†] | – | – | – | – |
| ConvNeXt-2.5-MIL | $-0.0094\,[-0.0172,\,-0.0029]$ | 0.0040 | $-0.0346\,[-0.0747,\,0.0057]$ | 0.0820 |
| Med3D-ResNet | $-0.4218\,[-0.4725,\,-0.3705]$ | $< 10^{-4}$ | $-0.7054\,[-0.7717,\,-0.6292]$ | $< 10^{-4}$ |
| ResNet-vanilla | $-0.2085\,[-0.2403,\,-0.1773]$ | $< 10^{-4}$ | $-0.6541\,[-0.7252,\,-0.5732]$ | $< 10^{-4}$ |

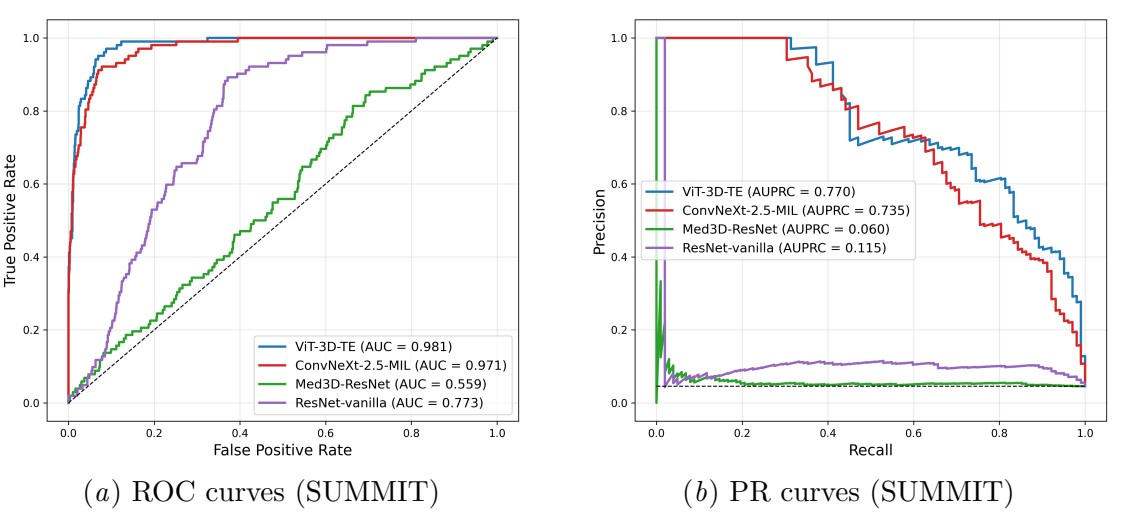

(a) ROC curves (SUMMIT)     (b) PR curves (SUMMIT)

Figure 4: Receiver operating characteristic (ROC) and precision-recall (PR) curves for the four main models on the SUMMIT internal test set. (a) ROC curves show the trade-off between sensitivity and specificity. (b) PR curves highlight performance under class imbalance.

Table 4: External validation on the COPDGene cohort. AUROC and AUPRC include 95% bootstrap confidence intervals.

| Model | AUROC (95% CI) | AUPRC (95% CI) | Sensitivity | Specificity |
|---|---|---|---|---|
| ViT-3D-TE | 0.9705 [0.9622, 0.9777] | 0.6170 [0.5535, 0.6771] | 0.9020 | 0.9119 |
| ConvNeXt-2.5-MIL | 0.9672 [0.9573, 0.9758] | 0.6051 [0.5405, 0.6672] | 0.9227 | 0.8876 |
| Med3D-ResNet | 0.5658 [0.5273, 0.6015] | 0.0313 [0.0254, 0.0379] | 0.7000 | 0.3981 |
| ResNet-vanilla | 0.7355 [0.7098, 0.7598] | 0.0471 [0.0394, 0.0558] | 0.9227 | 0.4571 |

Table 5: Paired bootstrap significance testing for the COPDGene external dataset. Baseline model is marked with †. $\Delta$ values denote paired differences relative to the baseline.

| Model | $\Delta$AUROC (95% CI) | p(AUROC) | $\Delta$AUPRC (95% CI) | p(AUPRC) |
|---|---|---|---|---|
| ViT-3D-TE† | – | – | – | – |
| ConvNeXt-2.5-MIL | $-0.0033$ [$-0.0090$, 0.0022] | 0.2630 | $-0.0119$ [$-0.0464$, 0.0250] | 0.5130 |
| Med3D-ResNet | $-0.4047$ [$-0.4423$, $-0.3680$] | $< 10^{-4}$ | $-0.5857$ [$-0.6448$, $-0.5229$] | $< 10^{-4}$ |
| ResNet-vanilla | $-0.2350$ [$-0.2591$, $-0.2120$] | $< 10^{-4}$ | $-0.5699$ [$-0.6287$, $-0.5072$] | $< 10^{-4}$ |

and patient characteristics (Table 4; Appendix Fig.6a–b). Despite this distribution shift, ViT-3D-TE maintained strong performance (AUROC 0.9705, AUPRC 0.6170), outperforming all comparison models.

ConvNeXt-2.5-MIL was again the closest competitor (AUROC 0.9672, AUPRC 0.6051), with differences versus ViT-3D-TE not reaching statistical significance ($\Delta$AUROC $= -0.0033$, $p = 0.263$; $\Delta$AUPRC $= -0.0119$, $p = 0.513$), indicating that the 2.5D slice-based MIL model retains reasonable robustness under distribution shift.

As in the internal evaluation, convolutional baselines showed substantial performance drops under cross-cohort shift: ResNet-vanilla and Med3D-ResNet achieved AUROCs of 0.7355 and 0.5658, with AUPRCs below 0.05, and were significantly inferior to ViT-3D-TE across all metrics (all $p < 10^{-4}$). ROC and PR curves (Appendix Fig. 6(a) a–b) show ViT-3D-TE preserving a stable margin across the operating range, ConvNeXt-2.5-MIL diverging in the low–false-positive regime, and convolutional baselines failing to achieve meaningful precision at COPDGene's low prevalence.

Overall, these results demonstrate that ViT-3D-TE generalises from SUMMIT to COPDGene despite substantial domain shift, underscoring the robustness of TANGERINE-based full-volume Transformer representations.

**Interpreting AUPRC in the context of disease prevalence.** Because AUPRC is strongly prevalence-dependent, it is informative to compare model performance against the expected precision–recall curve of a naïve classifier. In the SUMMIT internal test set, UILD prevalence was 4.6% (102/2237), corresponding to a random-guessing AUPRC of 0.046. ViT-3D-TE achieved an AUPRC of 0.770, representing a 17-fold improvement over this baseline.

The external COPDGene cohort exhibited even lower prevalence (2.5%; 220/8874), yielding a random AUPRC of 0.025. ViT-3D-TE achieved an AUPRC of 0.617, a 25-

fold improvement over random performance. These gains emphasise that the model offers substantial clinical value beyond what is attainable from prevalence-driven heuristics, particularly in large-scale screening settings where UILD is rare.

## 4.3. Architecture Ablation Study

Table 6: Ablation study on SUMMIT internal cohort evaluating variants of the ViT-based classification architecture. AUROC and AUPRC include 95% bootstrap confidence intervals.

| Model Variant | AUROC (95% CI) | AUPRC (95% CI) | Sensitivity | Specificity |
|---|---|---|---|---|
| ViT-3D-TE | 0.9805 [0.9709, 0.9875] | 0.7699 [0.6918, 0.8312] | 0.9706 | 0.9091 |
| ViT-HF | 0.9695 [0.9565, 0.9796] | 0.6943 [0.6119, 0.7731] | 0.8922 | 0.9330 |
| ViT-C | 0.9458 [0.9110, 0.9731] | 0.7522 [0.6715, 0.8196] | 0.9314 | 0.8084 |
| ViT-NP | 0.7625 [0.7132, 0.8100] | 0.1668 [0.1192, 0.2434] | 0.8725 | 0.4075 |

Table 7: Ablation study on the COPDGene external cohort. AUROC and AUPRC include 95% bootstrap confidence intervals.

| Model Variant | AUROC (95% CI) | AUPRC (95% CI) | Sensitivity | Specificity |
|---|---|---|---|---|
| ViT-3D-TE | 0.9705 [0.9622, 0.9777] | 0.6170 [0.5535, 0.6771] | 0.9020 | 0.9119 |
| ViT-HF | 0.9574 [0.9445, 0.9680] | 0.5814 [0.5161, 0.6454] | 0.8636 | 0.9078 |
| ViT-C | 0.9054 [0.8753, 0.9325] | 0.5687 [0.5006, 0.6327] | 0.9091 | 0.5484 |
| ViT-NP | 0.7543 [0.7235, 0.7825] | 0.0841 [0.0624, 0.1150] | 0.7500 | 0.6086 |

We conducted a series of ablations to isolate the contributions of key architectural design choices within the ViT-based classifier, examining (i) multi-token aggregation, (ii) hierarchical fusion, (iii) use of the CLS token alone, and (iv) removal of TANGERINE pretraining (Tables 6–7).

On the SUMMIT internal test set, ViT-3D-TE achieved the strongest overall performance (AUROC 0.981, AUPRC 0.770). ViT-C underperformed on AUROC (0.946) but achieved a surprisingly competitive AUPRC (0.752), indicating that the CLS token alone captures coarse-scale disease signal despite discarding patchwise spatial context. However, this CLS-only formulation yielded poorer thresholded performance, suggesting that although the CLS token preserves global structure for threshold-free metrics, it leads to unstable decision boundaries and systematic overcalling when a clinically constrained sensitivity operating point is imposed.

Under external distribution shift, the limitations of the CLS-only strategy became more pronounced. On COPDGene, ViT-C performance degraded (AUROC 0.905; AUPRC 0.569), whereas ViT-3D-TE generalised more strongly (AUROC 0.971; AUPRC 0.617). This suggests that triplet token aggregation, by preserving focal, diffuse, and global structure, improves robustness across scanners, kernels, and patient populations.

The remaining ablations reinforce these conclusions. Hierarchical fusion (ViT-HF) consistently underperformed relative to ViT-3D-TE, suggesting that concatenating intermediate features may introduce representational noise without improving discrimination, although factors such as layer selection and fusion strategy may also contribute. Removing TANGERINE pretraining (ViT-NP) led to severe degradation across all metrics (internal AUROC ~0.76; AUPRC ~0.17, with external AUPRC falling below 0.10), confirming that large-scale 3D pretraining is critical for stable optimisation.

## 5. Discussion

In this work, we present, to our knowledge, the largest deep learning–based cross-cohort generalisation study for UILD detection–a subtle but clinically consequential entity associated with accelerated lung function decline, respiratory morbidity, and increased mortality (Putman et al., 2016; Sanders et al., 2023). Across two of the largest thoracic CT cohorts worldwide, we show that these subtle parenchymal alterations–typically challenging for routine radiological detection–can be identified using a combination of multi-token 3D Vision Transformers and large-scale foundation model pretraining.

**Clinical implications for population screening.** Our findings have direct relevance for the evolving landscape of population CT imaging. National lung cancer screening programmes, such as the UK Targeted Lung Health Check and comparable US initiatives, already produce millions of LDCT scans each year (de Koning et al., 2020; Team, 2011; Cancer Research UK, 2024; Dickson et al., 2023). Although these programmes were designed to detect malignancy, they also offer an opportunity to identify preclinical interstitial lung disease–including UILD, which recent screening-cohort studies have shown to be both common and prognostically severe. In the SUMMIT study, individuals with UILD exhibited significantly higher rates of respiratory hospitalisation and mortality than CT-normal participants, despite having no prior ILD diagnosis (Putman et al., 2016). Moreover, the imaging features that define UILD–particularly early traction bronchiolectasis distributed across multiple lobes–have been validated as markers of early fibrotic remodelling, with strong associations with symptoms, physiology, and survival (Hunninghake et al., 2022).

Yet UILD remains systematically under-recognised (Hatabu et al., 2020). Standard ATS/Fleischner classifications were designed for earlier-generation CT imaging and tend to undercall early fibrosis, particularly because traction bronchiolectasis is excluded from their fibrosis definition and minimum-volume thresholds ($> 5\%$) overlook subtle disease. The system presented here illustrates how deep learning could support radiologists by flagging subtle fibrotic changes, reducing missed early disease, and enabling streamlined referral for confirmatory assessment or specialist review.

**Effectiveness of foundation model pretraining for 3D volumetric tasks.** Our results highlight the central importance of foundation model pretraining for full-volume 3D CT analysis. ViT-3D-TE, initialised with the TANGERINE encoder, achieved the strongest performance on both SUMMIT (internal) and COPDGene (external), despite extremely low disease prevalence–4.6% and 2.5%, respectively. Against random-guessing AUPRC baselines of 0.046 (SUMMIT) and 0.025 (COPDGene), ViT-3D-TE achieved roughly 17-fold and

25-fold improvements, indicating that the model learns genuinely disease-relevant texture signatures.

By contrast, the unpretrained 3D ViT variant (ViT-NP) was unable to converge to a meaningful solution. This highlights the difficulty of purely supervised optimisation on full $256^3$ volumes, where texture cues are subtle, imbalance is severe, and features of interest are sparsely distributed in peripheral lung regions. Large-scale pretraining on nearly 100,000 LDCT volumes equips the model with anatomically grounded features that stabilise optimisation and substantially enhance sensitivity to subtle interstitial changes.

**Qualitative behaviour and limitations.** Visual inspection of representative screening cases (Fig. 3), reviewed with a thoracic radiologist, suggests that ViT-3D-TE tends to perform more reliably in cases exhibiting subtle, spatially distributed fibrotic features, including faint reticulation and early traction bronchiolectasis across multiple lobes. Limitations are primarily observed in extremely mild peripheral disease near the limits of radiological detectability, as well as in scans dominated by confounding parenchymal abnormalities such as severe emphysema, which can obscure or mimic early fibrotic patterns. Notably, such confounders are common in the negative class and are well represented in the COPDGene cohort, providing a realistic test of model behaviour in emphysema-enriched screening populations. These behaviours reflect the inherent ambiguity of UILD in population screening and reinforce the intended role of the system as a screening-support and triage tool rather than a stand-alone diagnostic system. Finally, these limitations are consistent with the use of downsampled $256^3$ inputs, for which the most subtle subpleural abnormalities may fall below the effective spatial resolution.

**2.5D MIL as a strong alternative when 3D pretraining is unavailable.** The differing reliance on pretraining between full-volume ViTs and the slice-based MIL model highlights complementary inductive biases: while full-volume ViTs benefit strongly from large-scale pretraining to stabilise global self-attention, ConvNeXt-2.5-MIL leverages convolutional locality and slice-wise decomposition to reduce optimisation difficulty and achieve competitive performance with substantially weaker reliance on volumetric pretraining (although the 2D ConvNeXt backbone is initialised with standard masked-image pretraining). The ConvNeXt-2.5-MIL model achieved performance markedly higher than any conventional (non-pretrained) 3D architecture, demonstrating that reliable UILD detection is feasible even without access to 3D foundation models. Its strong performance underscores the practical value of 2.5D MIL-based approaches in settings where full-volume 3D pretraining is not available or difficult to obtain. As modern screening infrastructures expand globally–including sites with constrained computational resources–such alternatives are likely to be important for equitable deployment of early fibrosis detection (Bolón-Canedo et al., 2024; Besiroglu et al., 2024).

**Limitations and opportunities for future work.** Several limitations warrant discussion. First, models operated on downsampled $256^3$ inputs, a necessary compromise given the quadratic memory cost of Transformer attention. Downsampling may attenuate the fine-grained textural signatures that define UILD–such as subtle subpleural bronchiolectasis or faint reticulation–potentially constraining maximum achievable performance. Future work could explore multi-resolution hierarchies, axial–coronal–sagittal fusion, or longitudinal modelling to quantify disease progression.

Second, UILD represents just one early fibrotic phenotype. Other preclinical airway- or interstitial-disease signatures may be detectable using similar methods and could form the basis of more comprehensive "multi-phenotype" screening tools. Extending this work to quantify progression, characterise regional fibrosis patterns, and integrate clinical, functional, or genomic predictors of risk would further strengthen its clinical relevance. Fine-grained differentiation among specific interstitial subtypes and common fibrotic mimics (e.g. emphysema-related changes or non-fibrotic ILAs), as well as dedicated analysis of model sensitivity in marginal or borderline cases where initial radiologist disagreement exists, were beyond the scope of this screening-focused study and constitute important directions for future work.

Finally, real-world deployment studies are needed to assess prospective performance, workflow integration, and radiologist–AI interaction. Given UILD's prognostic consequences and under-recognition in routine practice, AI-assisted triage may be particularly impactful in enabling timely specialist review and enrolment into preventive trials before irreversible fibrosis develops (Martinez et al., 2022).

**Conclusion.** This study demonstrates that subtle early interstitial abnormalities–long overlooked in routine LDCT interpretation–can be detected reliably using a foundation-model–initialised 3D Transformer. Multi-token fusion enhances sensitivity to both focal and diffuse parenchymal texture changes, large-scale 3D pretraining provides essential anatomical priors, and 2.5D MIL offers a strong alternative when 3D foundation models are unavailable or impractical. Together, these findings establish a feasible and scalable pathway for incorporating early ILD detection into population CT screening workflows, supporting earlier intervention and enabling identification of high-risk individuals for disease-modifying therapies and clinical research.

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

## Acknowledgments

This research and JJ was funded in whole or in part by the Wellcome Trust [227835/Z/23/Z ] and the NIHR UCLH Biomedical Research Centre, UK. NM acknowledges funding from the UKRI Centre for Doctoral Training (CDT) in AI-enabled Healthcare Systems, supported by UKRI grant [EP/S021612/1]. The SUMMIT Study is funded by GRAIL LLC. through a research grant awarded to S.M.J. as Principal Investigator. S.M.J. is supported by CRUK programme grant (EDDCPGM00002), and MRC Programme grant (MR/W025051/1). SMJ receives support from the CRUK Lung Cancer Centre of Excellence (C11496/ A30025) and the CRUK City of London Centre, the Rosetrees Trust, the Roy Castle Lung Cancer foundation, the Longfonds BREATH Consortia, MRC UKRMP2 Consortia, the Garfield Weston Trust and University College London Hospitals Charitable Foundation. SMJ's work is supported by a Stand Up To Cancer-LUNGevity- American Lung Association Lung Cancer Interception Dream Team Translational Research Grant and Johnson and Johnson (grant number: SU2C-AACR-DT23-17 to S.M. Dubinett and A.E. Spira). Stand Up To Cancer is a division of the Entertainment Industry Foundation. Research grants are administered by the American Association for Cancer Research, the Scientific Partner of SU2C.This work was partly undertaken at UCLH/UCL who received a proportion of funding from the Department of Health's NIHR Biomedical Research Centre's funding scheme.

**SUMMIT Consortium** The authors thank the SUMMIT Consortium for their contribution to this work.

**SUMMIT Consortium Authors:** Sam M Janes[1], Jennifer L Dickson[1], Carolyn Horst[1], Sophie Tisi[1], Helen Hall[1], Priyam Verghese[1], Andrew Creamer[1], Thomas Callender[1], Ruth Prendecki[1], Amyn Bhamani[1], Chuen Khaw[1], Mamta Ruparel[1], Monica L. Mullin[1], Tanya Patrick[1], Allan Hackshaw[2], Anne-Marie Hacker[2], Esther Arthur-Darkwa[2], Samantha L Quaife[3], Arjun Nair[4], Anand Devaraj[5], Kylie Gyertson[4], Vicky Bowyer[4], Ethaar El-Emir[4], Judy Airebamen[4], Alice Cotton[4], Kaylene Phua[4], Elodie Murali[4], Simranjit Mehta[4], Janine Zylstra[4], Karen Parry-Billings[4], Columbus Ife[4], April Neville[4], Paul Robinson[4], Laura Green[4], Zahra Hanif[4], Helen Kiconco[4], Ricardo McEwen[4], Dominique Arancon[4], Nicholas Beech[4], Derya Ovayolu[4], Christine Hosein[4], Sylvia Patricia Enes[4], Jane Rowlands[4], Sheetal Karavadra[4], Aashna Samson[4], Urja Patel[4], Fahmida Hoque[4], Hina Pervez[4], Sofia Nnorom[4], Moksud Miah[4], Julian McKee[4], Mark Clark[4], Jeannie Eng[4], Fanta Bojang[4], Claire Levermore[4], Anant Patel[6], Sara Lock[7], Alan Shaw[7], Rajesh Banka[8], Angshu Bhowmik[9], Ugo Ekeowa[10], Chris Valerio[11], William M Ricketts[12], Neal Navani[4], Ali Mohammed[12], Terry O'Shaughnessy[12], Charlotte Cash[6], Magali Taylor[4], Samanjit Hare[6], Tunku Aziz[12], Stephen Ellis[12], Anthony Edey[13], Graham Robinson[14], Alberto Villanueva[15], Hasti Robbie[16], Elena Stefan[10], Charlie Sayer[17], Nick Screaton[18], Navinah Nundlall[4], Lynsey Gallagher[4], Andrew Crossingham[4], Thea Buchan[4], Tanita Limani[4], Kate Gowers[1], Kate Davies[1], John McCabe[1], Joseph Jacob[19], Mehran Azimbagirad[19], Burcu Ozaltin[19], Tania Anastasiadis[20], Andrew Perugia[21], James Rusius[21], Geoff Bellingan[4], Maureen Browne[4], Eleanor Hellier[4], Catherine Nestor[4].

**Affiliations:** (1) Lungs for Living Research Centre, UCL Respiratory, University College London, London; (2) CRUK & UCL Cancer Trials Centre, University College London, London; (3) Centre for Cancer Screening, Prevention, Detection and Early Diagnosis, Wolfson Institute of Population Health, Barts & The London School of Medicine and Dentistry, Queen Mary University of London, London; (4) University College London Hospitals NHS Foundation Trust, London; (5) RBHT, Imperial College; (6) Royal Free London NHS Foundation Trust, London; (7) Whittington Health NHS Trust, London; (8) Barking, Havering and Redbridge University Hospitals NHS Trust, Essex; (9) Homerton University Hospital Foundation Trust, London; (10) The Princess Alexandra Hospital NHS Trust, Essex; (11) North Middlesex University Hospital NHS Trust, London; (12) Barts Health NHS Trust, London; (13) North Bristol NHS Trust, Bristol; (14) Royal United Hospitals Bath NHS Foundation Trust, Bath; (15) Surrey and Sussex Healthcare NHS Trust, Surrey; (16) King's College Hospital NHS Foundation Trust, London; (17) University Hospitals Sussex NHS Foundation Trust, Sussex; (18) Royal Papworth Hospital NHS Foundation Trust, Cambridge; (19) Satsuma Lab, Centre for Medical Image Computing (CMIC), London; (20) Tower Hamlets Clinical Commissioning Group, London; (21) Noclor Research Support, London.

# Appendix A. Figures

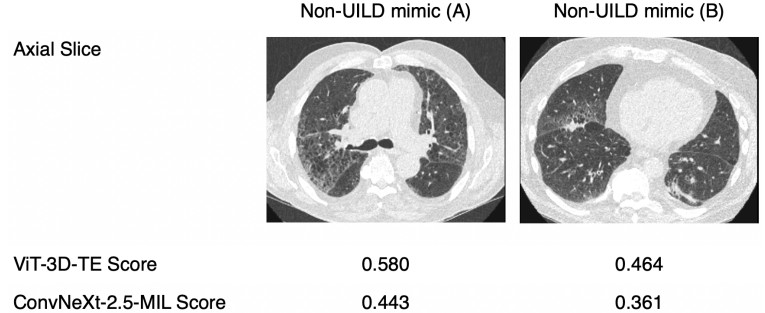

Figure 5: Non-UILD mimics illustrating confounding patterns. Representative non-UILD cases in which fibrotic-appearing parenchymal patterns elicit elevated UILD probability scores. (A) Extensive ground-glass opacities superimposed on emphysema, producing a coarse reticular appearance that may resemble honeycombing or fibrotic change. (B) Pleural plaques with associated band-like and rounded atelectasis, which can introduce linear subpleural opacities mimicking early fibrotic involvement. For each case, predicted UILD probabilities from ViT-3D-TE and ConvNeXt-2.5-MIL are reported. Images are cropped to the lung fields and displayed using a consistent lung window; all slices correspond to the $256^3$ down-sampled volumes used as model input.

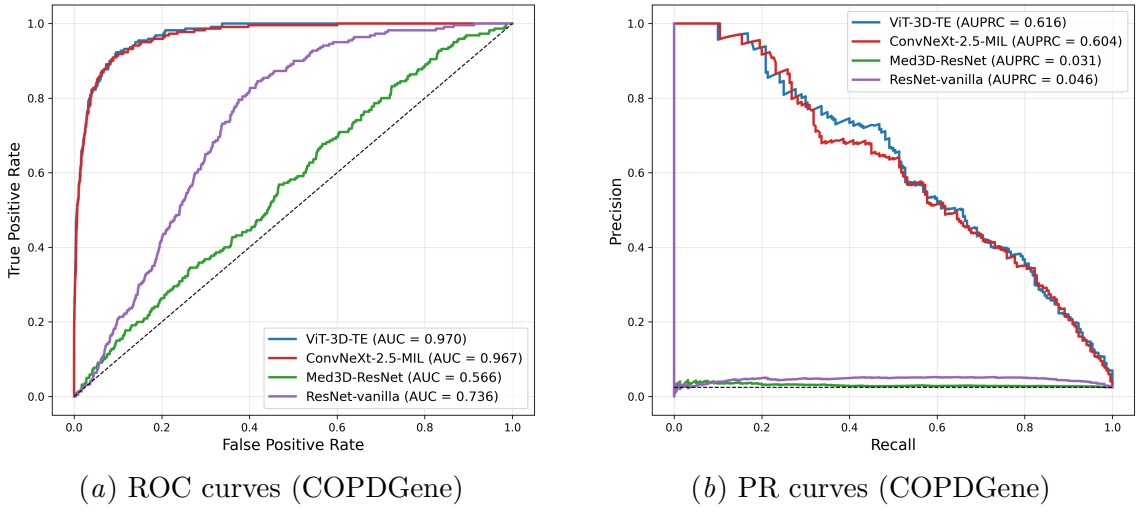

(*a*) ROC curves (COPDGene)   (*b*) PR curves (COPDGene)

Figure 6: Receiver operating characteristic (ROC) and precision-recall (PR) curves for the four main models on the COPDGene external test set. (a) ROC curves show the trade-off between sensitivity and specificity. (b) PR curves highlight performance under class imbalance.

