# OpenReview forum: "Scalable Detection of Undiagnosed ILD in Population Screening: A Multi-Cohort Study using 3D Foundation Models"
_MIDL.io/2026/Validation_Papers — MIDL 2026 - Validation Papers Poster_

### Official Review · Reviewer_orRG · 2025-12-31

**Confidence:** 3
**Preliminary Rating:** 4
**Final Rating:** 4

**Summary:**

→This paper proposes an automated method for detecting undiagnosed interstitial lung disease (UILD) in large-scale low-dose CT (LDCT) population screening and evaluates it systematically on two of the largest chest CT cohorts worldwide: the UK SUMMIT cohort (over 11,000 LDCT scans) and the U.S. multi-center COPDGene cohort (over 8,800 scans acquired from 21 scanners). The authors introduce ViT-3D-TE, a multi-token fusion architecture based on a 3D Vision Transformer that jointly integrates CLS, MAX, and AVG tokens to preserve global thoracic context, localized high-frequency texture cues, and diffuse low-contrast parenchymal changes, enabling improved capture of early fibrotic patterns. The model is initialized with a large-scale 3D foundation model (TANGERINE) and demonstrates stable performance that is consistently superior to conventional 3D CNNs and non-pretrained Transformers under extremely low disease prevalence conditions. In addition, the authors propose ConvNeXt-2.5-MIL as a lightweight alternative, which leverages slice-level multiple instance learning and craniocaudal positional encoding to achieve competitive performance without 3D pretraining, addressing resource-constrained scenarios. Overall, the results suggest the potential feasibility and clinical value of integrating AI-based systems into national LDCT screening workflows for early ILD identification.

**Strengths:**

1. The paper focuses on the detection of undiagnosed interstitial lung disease (UILD) in LDCT population screening, a problem that has been relatively underexplored compared to studies centered on already diagnosed ILD.
2. The study is validated on two of the largest chest CT cohorts available.
3. Clear motivation and targeted improvements in the ViT-3D-TE design.

**Weaknesses:**

1. All models are trained on volumetric data downsampled to a resolution of 256³, which may reduce the ability to capture extremely subtle textural features, such as subtle subpleural bronchiolectasis or faint reticulation. Although this limitation is acknowledged in the discussion, the quantitative impact of this design choice on model performance is not analyzed, leaving the associated trade-offs insufficiently clear.
2. While the detection performance is competitive, the paper does not provide attention maps, saliency visualizations, or case-level examples. As a result, it is difficult to verify whether the model focuses on imaging patterns relevant to UILD rather than scanner artifacts or unrelated structures. This absence may limit clinical confidence in the model’s decision-making process in a screening context.
3. The paper provides a thorough discussion of the limitations of commercial systems such as ScreenDx in terms of evaluation protocols and reported metrics, which objectively restricts the feasibility of direct comparison under a unified framework. However, in the absence of aligned quantitative results, it remains difficult for readers to assess the magnitude of the proposed method’s performance gain over existing clinical tools in practical screening workflows.
4. Collapsing all non-UILD cases into a single negative class helps simplify the modeling objective but may obscure the model’s ability to distinguish among different types of interstitial abnormalities or common confounding conditions, thereby limiting the granularity of clinical interpretation.

**Detailed Comments:**

1. Adding interpretability results, such as overlap analysis between model attention maps and radiologist-annotated lesion regions, would help validate the clinical plausibility of the model’s decisions.
2. It may be beneficial to include ablation experiments at different input resolutions (e.g., 128^3, 256^3, and 512^3), or to compare model performance between subgroups with pronounced UILD and those with very mild UILD, in order to more clearly delineate the scope of the impact of the downsampling strategy.
3. It would be helpful to clarify whether the performance of ConvNeXt-2.5-MIL is sensitive to the choice of slice number (e.g., N = 128), either through brief justification or additional experimental results.

**Justification Of Final Rating:**

The authors have addressed the reviewer's concerns by adding interpretability analyses, clarifying the impact of downsampling, and explaining architectural differences. Although the visualization results are still limited of interpretability, the revisions improve the manuscript's clarity and clinical relevance. Therefore, I maintain a weak accept rating.

**Justification Of The Preliminary Rating:**

Overall, this paepr presents a reasonable and potential impactful approach, but issues like resolution sensitivity, interpretability, and clinical positioning remain insufficiently addressed. For these reasons, I assign a weak accept rating.

**Questions To Address In The Rebuttal:**

1. Ablation results show a substantial performance drop for ViT-3D-TE when TANGERINE pretraining is removed, whereas ConvNeXt-2.5-MIL achieves relatively comparable performance without 3D medical image pretraining. Can the authors further explain the differences in inductive bias and optimization stability between these architectures, and how these differences manifest in the UILD detection task?
2. The paper reports that ViT-HF underperforms ViT-3D-TE and attributes this to representational noise introduced by intermediate features. Could alternative factors—such as layer selection, channel alignment, or fusion strategy—also contribute to this outcome, and were other hierarchical configurations explored?
3. While downsampling to 256^3 improves computational efficiency, its effect on detecting very subtle UILD features remains unclear. Have the authors observed any performance differences between cases with pronounced UILD and those with very mild UILD, or have they evaluated the potential benefits of using higher-resolution inputs?

---

### Official Review · Reviewer_XFn4 · 2026-01-09

**Confidence:** 5
**Preliminary Rating:** 5
**Final Rating:** 5

**Summary:**

This paper addressed Scalable Detection of Undiagnosed Interstitial Lung Disease (UILD) in Population Screening: A Multi-Cohort Study using 3D Foundation Models. The paper developed and validated a foundation-model–augmented deep learning system for UILD detection across two of the largest thoracic CT cohorts worldwide: SUMMIT, and COPDGene. In addition, the paper proposed ViT-3D-TE, a multi-token 3D Vision Transformer that aims to preserve both high-frequency focal texture and diffuse parenchymal change through CLS, MAX, and AVG token fusion. The model was initialized with TANGERINE.  ViT-3D-TE. The results revealed that foundation model–enhanced 3D Transformers offer a practical and scalable pathway for integrating UILD detection into national LDCT screening workflows.

**Strengths:**

i. One of the strength of the paper was the utilization of world's largest thoracic CT cohorts (SUMMIT and COPDGene), totaling over 20,000 scans.
ii. Methodologically, the inclusion of ConvNeXt-2.5-MIL as a competitive baseline was another strength as this provided an alternative for researchers who may not have access to 3D foundation models.

**Weaknesses:**

Inconsistence of tense usage. i.e . in abstract , "We develop and validate (Present Tense). While, in the Methods section shifts to "All CT scans underwent (Past Tense). More so,  In Section 3.4.3, it says "ConvNeXt-2.5-MIL... operates in a 2.5D regime" (Present). while in Section 3.4.4, it says "We adapted a 3D ViT-Large architecture" (Past).

**Detailed Comments:**

The present tense should be used to describe the model's nature, while past tense should be used to describe the particular steps the researchers did.

**Justification Of Final Rating:**

The authors have addressed all the concerns I raised during the review stage. All questions have been adequately answered, and additional figures were provided to further support and justify their work. Based on this, I am recommending a strong accept.

**Justification Of The Preliminary Rating:**

The combination of technological innovation and extensive clinical validation justified the rating of 5 (Strong Accept). Additionally, the authors used a unique 3D Vision Transformer ensemble (ViT-3D-TE) to address Undiagnosed Interstitial Lung Disease, a high-impact "blind spot" in radiology. The results were statistically accurate, and the study used a good methodological approach.

**Questions To Address In The Rebuttal:**

1. Did the authors carefully assess the sensitivity of the model on the "marginal" scenarios when the consensus radiologists first couldn't agree?
2. How does the model function when heavy "mimics" that were part of the wide negative class, like severe emphysema or acute infection (pneumonia) are present?

---

### Official Review · Reviewer_yNUN · 2026-01-09

**Confidence:** 5
**Preliminary Rating:** 3
**Final Rating:** 4

**Summary:**

This paper presents a large-scale, multi-cohort study for detecting undiagnosed interstitial lung disease (UILD) from low-dose CT screening scans. The authors evaluate several deep learning models, including a proposed multi-token 3D Vision Transformer (ViT-3D-TE) and a 2.5D MIL-based ConvNeXt variant, leveraging large-scale foundation model pretraining and external validation across two data cohorts.

**Strengths:**

- The focus on UILD detection in population LDCT screening clearly addresses a genuine unmet need.

- Training on SUMMIT and external evaluation on COPDGene without domain adaptation.

- Quantitative evaluation protocol is reasonable.

**Weaknesses:**

- Despite strong quantitative performance, the paper provides very minimal qualitative visualization of model predictions

- The paper introduces ViT-3D-TE and describes it as a proposed architecture, yet the distinction between this design and existing token aggregation or pooling strategies is not always clear

**Detailed Comments:**

- The paper would benefit from more extensive qualitative examples showing both successful and failure cases. In particular, side-by-side visual comparisons between ViT-3D-TE and competing methods on challenging examples would help clarify whether the reported performance gains translate into clinically meaningful improvements.

- While the multi-token aggregation strategy is intuitively appealing, the paper does not clearly explain how it differs from existing pooling or multi-scale aggregation approaches, or why it should be considered a distinct architectural contribution rather than an implementation-level variant.

- Please more emphasis on interpretability and failure analysis beyond aggregate metrics.

**Justification Of Final Rating:**

The rebuttal addresses the main concerns in a concrete and satisfactory way. The added qualitative figures, including side-by-side comparisons and explicit failure cases reviewed by a radiologist, meaningfully improve interpretability beyond aggregate metrics. The clarification of ViT-3D-TE also helps position it as a task-driven architectural adaptation rather than an unspecified pooling variant.

However, the qualitative analysis remains illustrative rather than comprehensive, and the architectural contribution is better viewed as a practical refinement motivated by the screening task than a broadly general modeling approach.

**Justification Of The Preliminary Rating:**

The quantitative evaluation is extensive and carefully conducted, aspects of the qualitative validation and architectural positioning would benefit from greater clarity, particularly given the paper’s emphasis on robustness and translational relevance.

**Questions To Address In The Rebuttal:**

Please check the weakness and detailed comments sections.

---

### Author Rebuttal · Authors · 2026-01-24

**Rebuttal:**

We thank the reviewers for their careful reading of the manuscript and for the constructive and insightful feedback provided. We believe that the main concerns raised across all reviews have now been fully addressed.

In particular, we have strengthened the qualitative validation and interpretability of the study through additional case-level visualisations, clarified the architectural positioning and motivation of ViT-3D-TE, ensured consistent tense usage throughout the manuscript, and expanded the discussion of model behaviour in borderline cases and in the presence of common confounding mimics. We have also further clarified the scope, limitations, and intended screening-oriented role of the proposed system.

Please find the revised manuscript attached, with all changes highlighted in red. We hope that the revisions satisfactorily address the reviewers’ comments and improve the clarity, interpretability, and clinical framing of the work.

**Supporting Material:**

/attachment/eb3fcb0ee5e1e608663d9c4333a33f95e03f6f35.pdf

---

### Meta-Review · Area_Chair_53uX · 2026-02-09

**Recommendation:** Accept (Poster)
**Confidence:** 4

**Metareview:**

All reviewers converge on an accept recommendation after the rebuttal. The rebuttal directly addresses the main shared concerns, including qualitative interpretability and clinical plausibility, clearer positioning of the ViT-3D-TE design, and tighter clinical framing for population screening. Overall, the revised manuscript supports the claims and provides a useful and clinically relevant validation study for the community.

---

### Decision · Program_Chairs · 2026-02-14

Accept (Poster)